# Productive and reproductive performance of Dawuro highland sheep managed under community-based breeding program in Ethiopia

**Kebede Habtegiorgis Beshah**[1]*, **Adisu Jimma**[1], **Deribe Gemiyo**[1], **Ayele Abebe**[2]

**1** Southern Agricultural Research Institute, Areka Agricultural Research Center, Areka, Ethiopia, **2** Debre Birhan Agricultural Research Center, Deber Berhan, Ethiopia

* kebede.habtegiorgis@gmail.com

**Data Availability Statement:** The data underlying the results presented in the study are owned by Ethiopian institute of agriculture research (EIAR),

## Abstract

The objective of this study was to evaluate growth and reproductive performance of highland Dawuro sheep population managed under a community-based breeding program in Ethiopia. Records used in the study were collected over 6 years (2017–2022). In the current study, 3554 records were considered. Studied traits were birth weight (BWT), 3-month weight (WWT), 6-month weight (SMWT), daily gain from birth to weaning (ADG0-3), daily gains from weaning to 6-month age(ADG3-6), litter size(LS), lambing interval (LI), and annual reproductive rates(ARR). The data were analyzed using general linear models of SAS software. The overall least squares mean (LSM± SE) for BWT, WWT, SMWT, ADG0-3, and ADG3-6 were 2.69±0.02; 14.6±0.49, 22.8±0.22 kg, 128.05±2.27 gram (g), and 71.5 ±0.23 gram respectively. Site difference exerted significantly on BWT, WWT, and SMWT. The result of WWT and SMWT results of Dawuro sheep were comparable with previously reported known Ethiopian sheep breeds. The overall least squares mean ± standard error of LS, LI and, ARR were 1.67±0.04 lambs, 239±31.91 days, and 2.19±0.06 lambs, respectively. WWT and SMWT traits showed good responses to selection. The general implication of the result to future improvement program was continue selection, strengthening the existing breeder cooperatives, integration of full animal health, feed packages, conservation of prolific flocks, improving the management of animals, expanding the current CBBP to adjacent potential kebele, create market linkage and scaling out the existing best practice to a new sites.

## Introduction

Sheep production is one of the major parts of the livestock production system in Ethiopia. They are widely reared in crop-livestock farming systems and are distributed across different agro-ecological zones of the country. According to [1], the estimated sheep population in Ethiopia was 42.9 million. Of this figure, 99.52 percent were indigenous breeds. With these huge

Agricultural growth program(AGP II), Southern agricultural research institute (SARI), and Areka agricultural research centre. Accordingly, data are available from the Ethiopian EIAR (www.eiar.gov. et) from the Livestock Research directorate, Tell: 0116-45 44 32/ Or 0913380858/0116457412, email address: livestock.research@eiar.gov.et. In addition, data are available from the southern agricultural research institute(www.sari.gov.et) livestock research directorate, South Ethiopia, Hawasa via email: livstokresearch@sari.gov.et.

**Funding:** This study was supported by Ethiopian Agricultural Research Institute (grant numbers: 31-20-05), Agricultural growth program https://www. gafspfund.org/projects/agricultural-growth-project-ii-agp-ii / and Southern Agricultural Research Institute, Areka agricultural research center.

**Competing interests:** The authors have declared that no competing interests exist.

**Abbreviations:** ADG0-3, daily gain from birth to weaning; ADG3-6, daily gains from weaning to 6-month age; ARR, annual reproductive rates; BWT, Birth weight; CBBP, community based breeding program; EIAR, Ethiopian Institute of Agricultural Research; ICARDA, international centre for agricultural research in the dry areas; LI, lambing interval; LS, litter size; LSM, least squares means; m.a.s.l, meter above sea level; N, number of observation; SAS, statistical analysis system; SE, Standard errors; SMWT, 6-month weight; SNNPR, southern nation nationality and people regional government; WWT, 3-month weight.

resources and because of various other reasons, sheep production has advantageous compared to other livestock production. However, production was in traditional way with high lamb mortality [2]. Therefore, an integrated attempt with emphasis on genetic improvement is crucial for improving animal output by improving growth and reproductive traits [3].

As indicated by [2–5], CBBPs are being implemented in various communities in Ethiopia intended to increase sheep productive and reproductive performances. Evaluation of reproductive and productive performance for the native indigenous sheep population is essential for conservatory considerations [6], to test the efficiency of applied CBBP [7] and to provide breeder, researcher and policy maker with information to develop more efficient selection program in the future [4, 7]. However, studies on the performance of CBBPs on productive and reproductive traits in sheep are scarce in Ethiopia [4].

Dawuro sheep CBBP is established in 2016/17 in Dawuro zone of Tocha and Kechi Tuta districts. Dawuro sheep were known for their fattening potential, twining rate, short lambing interval, and resistance to disease [8–10]. According to the report of [11], Dawuro sheep population are resembles with Bonga sheep breed. There is a data collected since the CBBP establishments. The data recorded contain productive and reproductive records. Reproductive and productive performance for Dawuro sheep under CBBP is unavailable. Few preliminary works with small data set were conducted on growth performance of Dawuro sheep by [12] and production system by [13]. Thus, it is important to evaluate growth and reproductive performance of Dawuro sheep in the on-going breeding programs. Moreover, evaluation of performance traits gives an indication of genetic improvement direction of the sheep population as well as the rate of important traits improvement from the time of application of the breeding program [7, 13]. Accordingly, the present study has been planned to evaluate the data generated under this program. Thus, there was a need to evaluate the data generated on the ongoing data collection to know the genetic improvement trend of this sheep type under the CBBP.

## Materials and methods

### Ethics statement

Data collection formats and procedures were prepared by international center for agricultural research in the dry areas (ICARDA). The studied animals are generally managed and owned by CBBP participant farmers and data were collected with their permission. No animals were injured during performance data collection. CBBP participant farmers provided their verbal informed consent to participate in this study. Participant farmer's ethnicity and their religious issues were not asked or recorded during data collection.

### Description of the study area

The study area (Dawuro sheep CBBP) was found in the Dawuro Zone of southern Ethiopia at a distance of 507 km to the Southwest of Addis Ababa national capital. Dawuro is situated at an altitude ranging from 800 to 2850 m.a.s.l., longitude 37˚09'E and latitude 7˚08 'N. The capital of the Dawuro zone is Tarcha, which is located at about 507 km from Addis Ababa and 282 km from Hawassa (the capital city of SNNPR).

The annual mean maximum and minimum temperatures of the Dawuro zone are 18˚C and 29˚C, respectively (http://climexp.knmi.nl/select.cgi?field=cru4_tmp/). The annual mean rainfall of the zone ranged from 1200 to 1800 mm (http://climexp.knmi.nl/select.cgi?field=cru4_ pre). The main rainy season of the zone is from June to September (the long rainy season), the short rainy season from March to April, and the dry season lasts from October to February and May. Agro ecology of study area classified as highland ≥ 2300 m.a.s.l., 44.1%, midland

>1500 to < 2300 m.a.s.l., 19.5% of area; and lowland ≤ 1500 m.a.s.l., 36.4% of area. Altitude ranges between 800 to 2850 m.a.s.l.

## Breeding program and animal management

Animals were identified by plastic ear tags. In each of the breeder cooperatives, an enumerator was employed for routine animal identification, data recording, and follow-up. Enumerators use herd books for data recording. The selection of breeding rams takes place on a programmed date, twice per year. Selected best breeding rams usually serve for two years in the community flock. Flocks were kept in animal houses during the night and midday. The houses were made of bamboo walls corrugated with any locally available roofing materials. The main feed sources for animals included natural pasture, crop residues and kitchen leftovers. Flocks grazing with tethering on the small private land was common practice in three study areas. Feed availability and abundance vary with rainfall patterns, among sites and owners. Comparatively huge amount of feed resources were available in the rainy season whereas less in quality and quantity during the dry season.

## Data sets

Data used for the study were obtained from the ongoing three breeder cooperatives. The performance data along with pedigree information is being maintained in the data-recording book of individual breeder cooperatives. The data routinely collected by the enumerators were recorded at the time of the event. The birth weight (BWT) was recorded within 24 hours of lambing; weaning weight (WWT) was taken from 90 days of age, and 6-month weight (SMWT) was taken from 180 days of age. The average daily body weight gains from birth to weaning age; and weaning to six months of age has been estimated as under:

$$\text{daily BW gain up to weaning age (g)} = \frac{\text{WWT} - \text{BWT}}{90} * 1000 \qquad (1)$$

$$\text{daily BW gain from weaning up to 6 months age (g)} = \frac{\text{SMWT} - \text{WWT}}{90} * 1000 \qquad (2)$$

Where:
BW = body weight
BWT = Birth weight,
WWT = Weaning weight at 90 days,
SMWT = 6-month weight at 180 days
The annual reproductive rate (ARR) for Dawuro ewes was calculated by the following formula adopted by [14]

$$ARR = \frac{365 * \text{average litter size}}{\text{average days of lambing interval}} \qquad (3)$$

## Statistical analysis

Data used for analysis included birth weight, three-month weight, 6-month weight, lambing interval, liter size, and ARR. Before conducting the main analysis, data were checked for homogeneity and normality test. Data were analyzed using the Generalized Linear Model (GLM) procedures of SAS version 9.2 (SAS 9.2, 2008). The factors used in the model included a year of birth/lambing (2013 to 2018), seasons (main rainy seasons, small rainy and dry seasons), sex (male and female), parity (1, 2, 3, 4, 5, 6 and ≥7), birth type (single, twin, and triplet or above) and site (Kechi, Medeaniyalem and Gebera). Tukey–Kramer test was used to

separate least-squares means with more than two levels. The maximum number of ewes parities for the data were 10.

**The model for growth traits and daily weight gain traits was as follows,**

$$Y_{ijklmno} = \mu + P_i + S_j + B_k + Yr_l + SE_m + SX_n + OWo(Yr_i * SE_m)_p + (B_k * Yr_l)_q$$
$$+ e_{ijklmnopq} \qquad \text{(Model 1)}$$

Where:

$Y_{ijklmno}$ = growth trait for each animal

$\mu$ = overall mean,

$P_i$ = $i^{th}$ parity (i = 7; 1, 2, 3, 4, 5, 6, $\geq$7)

$S_j$ = $j^{th}$ site (j = 5; Kechi, Medeaniyalem and Gebera)

$B_k$ = $k^{th}$ birth type (k = 3; single, twin, triplet and above)

$Yr_l$ = $l^{th}$ year (l = 6; 2017–2022)

$SE_m$ = $m^{th}$ season (m = 3; main rainy season, Small rainy, dry season)

$SX_n$ = $n^{th}$ sex (n = male, female)

$OW_o$ = $o^{th}$ Animal owner(o = Animal owner list)

$Yr_i * SE_m$ = the interaction between year and season of lambing;

$B_k * Yr_l$ = the interaction between birth type and year of lambing and

$e_{ijklmnopq}$ = random error

**The model for Lambing interval, Litter size, and ARR was as follows,**

$$Y_{ijklmno} = \mu + P_i + S_j + B_k + Y_l + SE_{m+} OWn + (Y_i * SE_m)_o + e_{ijklmno} \qquad \text{(Model 2)}$$

Where:

$\mu$ = overall mean,

$P_i$ = $i^{th}$ parity (i = 7; 1, 2, 3, 4, 5, 6, $\geq$7)

$S_j$ = $j^{th}$ site (j = 5; Kechi, Medeaniyalem and Gebera)

$B_k$ = $k^{th}$ birth type (k = 3; single, twin, triplet and above)

$Y_l$ = $l^{th}$ year (l-6; 2013–2018)

$SE_m$ = $m^{th}$ season (m = 3; main rainy season, Small rainy, dry season)

$OW_n$ = $n^{th}$ Animal owner(n = Animal owner list)

$Y_l * SE_m$ = the interaction between year and season of lambing

$e_{ijklmno}$ = random error

Notice: The fixed effect birth type was not used for Litter size

## Results and discussion

### Growth performance

**Birth weight (BWT).** The overall least square mean of BWT was 2.69±0.02 kg and all considered fixed effects were significant at P<0.01(Table 1). The interaction effect between season with year and lamb birth type by year were significant. There was a difference in the LSM±SE of BWT among ewe's parties. The heavier BWT was recorded from parity≥7, which was 2.75 ±0.20 kg. A smaller mean of BWT (2.62±0.02 kg) was observed in Gebera site. This reason could be due to the variation in the environment or management of animals. Additionally, Gebera site is established relatively earlier than the two sites.

The value of means among single, twin and triplet and above born lambs were 2.95±0.02 kg, 2.63±0.02 kg, and 2.42±0.05 kg respectively. The BWT of male lambs was found significantly heavier than their female counterparts (2.72±0.02 kg vs 2.62±0.02 kg).The smaller mean of BWT 2.65±0.03 was recorded in 2018 and it could be due to the variation in the

**Table 1. Least squares means (LSM±S.E) for growth traits of Dawuro sheep.**

| Source of Variation | BWT(Kg) | | WWT(Kg) | | SMWT(Kg) | |
|---|---|---|---|---|---|---|
| | N | mean ±SE | N | Mean ±SE | N | LSM ±SE |
| Overall | 3508 | 2.65±0.02 | 2064 | 14.6±2.49 | 903 | 20.65±0.22 |
| CV% | | 13.63 | | 14.84 | | 16.00 |
| **Parity** | | < .0001 | | NS | | NS |
| Parity 1 | 700 | 2.58±0.02[c] | 517 | 13.84±0.09 | 269 | 20.84±0.20 |
| Parity 2 | 984 | 2.60±0.02[bc] | 625 | 13.99±0.08 | 271 | 20.83±0.20 |
| Parity 3 | 696 | 2.68±0.02[ab] | 381 | 14.06±0.11 | 141 | 20.75±0.28 |
| Parity 4 | 525 | 2.71±0.02[a] | 238 | 14.00±0.14 | 72 | 20.87±0.39 |
| Parity 5 | 268 | 2.71±0.02[abc] | 132 | 14.16±0.19 | 69 | 19.89±0.40 |
| Parity 6 | 140 | 2.64±0.03[abc] | 76 | 14.05±0.25 | 29 | 20.33±0.61 |
| Parity ≥7 | 194 | 2.75±0.03[a] | 95 | 14.40±0.22 | 52 | 21.05±0.46 |
| **Cooperative** | | < .0001 | | 0.004 | | 0.004 |
| Medhaniyalem | 918 | 2.69±0.03[a] | 745 | 14.03±0.08[d] | 542 | 21.09±0.16[a] |
| Kechi | 896 | 2.69±0.02[a] | 1278 | 14.34±0.06[c] | 361 | 20.21±0.13[b] |
| Gebera | 512 | 2.62±0.02[b] | 41 | 13.83±0.34[a] | - | - |
| **Birth type** | | < .0001 | | < .0001 | | 0.0002 |
| Single | 1209 | 2.95±0.02[a] | 726 | 15.32±0.08[a] | 529 | 21.57±0.18[a] |
| Twin | 2150 | 2.63±0.02[b] | 1263 | 14.20±0.06[b] | 712 | 20.59±0.14[a] |
| ≥Triplet | 148 | 2.42±0.05[c] | 75 | 12.69±0.25[c] | 63 | 19.79±0.54[b] |
| **Sex** | | < .0001 | | < .0001 | | < .0001 |
| Male | 1813 | 2.72±0.02[a] | 955 | 14.52±0.06 | 921 | 21.60±0.15[a] |
| Female | 1694 | 2.62±0.02[b] | 1109 | 13.62±0.07 | 383 | 19.71±0.16[b] |
| **Season** | | 0.01 | | 0.002 | | 0.052 |
| Rainy season | 1170 | 2.64±0.04[b] | 694 | 13.77±0.08[b] | 383 | 20.71±0.17 |
| Small rainy | 1240 | 2.71±0.02[a] | 653 | 14.25±0.08[a] | 365 | 20.24±0.17 |
| Dry season | 1197 | 2.64±0.02[b] | 717 | 14.18±0.08[b] | 155 | 21.01±0.26 |
| **Birth year** | | < .0001 | | 0.003 | | < .0001 |
| 2017 | 196 | 2.74±0.08[a] | 82 | 13.39±0.24[b] | - | - |
| 2018 | 899 | 2.54±0.02[b] | 412 | 13.55±0.10[b] | 191 | 18.04±0.24[d] |
| 2019 | 697 | 2.70±0.02[a] | 617 | 14.12±0.08[a] | 331 | 20.26±0.18[a] |
| 2020 | 628 | 2.60±0.03[b] | 509 | 14.09±0.09[b] | 199 | 22.11±0.23[bc] |
| 2021 | 847 | 2.70±0.02[b] | 423 | 14.28±0.10[c] | 182 | 22.20±0.24[c] |
| 2022 | 240 | 2.63±0.10[b] | 21 | 15.00±0.47[b] | - | - |
| Owners | | < .0001 | | < .0001 | | < .0001 |
| Season*year | | 0.0014 | | 0.02 | | < .0001 |
| Birth type*year | | < .0001 | | < .0001 | | NS |

Mean values with different superscripts ([a,b]) across columns are significantly different (P<0.05), LSM-Least Squares Means, SE-Standard Error, BWT-birth weight, WWT-weaning weight, SMWT-six-month weight, kg-kilograms, N-number of observations

environment or management over the years. The current mean BWT of Dawuro sheep(2.69 ±0.02 Kg) was comparable with the report of [15] which is 2.69±0.02 kg for Washera sheep breed under farmer management system. However, much higher BWT for Rutana sheep (3.12 ±0.13 Kg) under on farm management system was reported by [16]. In a similar study by [7] found BWT of 3.12 kg for Doyogean sheep under CBBP. [17] also reported heavier than this finding for Bonga sheep breed under CBBP. The difference of BWT among the breeds could be due to genetic difference of the breed.

**Weaning weight (WWT).** The overall least-squares means of WWT was 14.6±2.49. Except for ewe parity the effect of site, birth type, year, season, sex, interaction between season by year and lamb birth type by year were significant(P<0.01) on weaning weight of Dawuro sheep. Single-born lambs (15.32±0.48 kg) was heavy than a twin (14.20±0.02) and triplet or above triplet (12.69±0.25). This effect is attributed mainly because single lambs do not have to compete for nutrients, unlike what happens when multiple lambs were developed [18]. Male lambs were significantly heavy than female lambs. The differences in birth weights observed between the sexes might be due to the difference in testosterone secretion between males and females [15].

Lambs wean in the time of small shower rainfall were significantly (P = 0.001) heavy than those born in the main rainy season and dry seasons. The effect of the season is associated with the difference in feed and disease situation [18]. The heavy WWT during small rainfall was attributed to the presence of feed due to the short rainy season starting February to May. Birth year is a significant source of variation for WWT where the heavy WWT was recorded in 2022 while the lowest was at the start of CBBP in 2013.

WWT trend across years was relatively increasing trend. This could be due to the combined result of selection with better feed availability in the studied area. Results obtained in the current study were nearly comparable with the report of [7]; which is 14.8±2.49 kg for Doyogena sheep, under the same CBBP management system, [6]; 14.40±0.23kg for Rutana sheep under on farm management system, and [19]; 14.8±0.22kg for Bonga sheep breed under CBBP management system. However higher result of WWT 15.5 kg was reported by [17] for the Bonga sheep breed.

**Six-month weight (SMWT).** The overall least-squares mean of SMWT was 20.65±0.22 kg. The effect of site, birth type, year of birth, and sex, interaction between season with year and lamb birth type by year were significant at P<0.01 whereas the effect of parity and season were found to be non-significant (Table 1). The least-squares means of SMWT for single, twin, and triplet and above triplets' births were 21.57±0.18, 20.59±0.14, and 19.79±0.54 kg respectively. Males were superior to females 21.60±0.15 Vs 19.71±0.16kg. The results are relatively smaller than previous studies by [7] for Doyogena sheep and other several authors finding [6]. Birth year was a significant source of variation for SMWT where the heavy SMWT was recorded in 2021 while the lowest was in 2018.SMWT trend across years was in increasing. The SMWT (20.65±0.22 for kg) Dawuro sheep was lower than SMWT of Bonga sheep breed (22.2±0.21 kg; [17] Doyogena sheep breed (22±0.22; [7] and higher than (21.0±0.70; [19] under CBBP and higher than SMWT of Menz and Horro sheep breeds under CBBP [19]. The result suggested that wide variability exists among Ethiopian sheep breeds with respect to potential growth performance. SMWT of Dawuro sheep indicates the potential for meat production for the local market and export market for this sheep type within 6-month age.

## Average daily gain

**Pre-weaning average daily gain (ADG0-3).** The overall least squares mean of ADG0-3 was 128.05±2.27 gram /day. The effect of site, birth type, sex and year on ADG0-3 had significant (P<0.01) whereas the effect of parity and season were found to be non-significant (Table 2). The least-squares means of ADG0-3 for single, twin, and triple and above birth lambs were 138.13±0.87, 129.37±0.66 and 116.64±2.72 gram /day respectively. The average daily weight gains of triple and above born lambs were lower than single and twin types of births. Male lambs had higher ADG0-3 than females (132.53±0.70 vs 123.56±0.76 gram/day).

Year was a significant source of variation for ADG0-3 and the trend across years was increasing where the heavy ADG0-3 was recorded in 2022 while the lowest was in 2017. The

**Table 2. The least-squares mean (LSM±S.E) for daily weight gain traits.**

| Source of Variation | ADG0-3(g) | | ADG3-6(g) | |
| --- | --- | --- | --- | --- |
| | N | mean ±SE | N | mean ±SE |
| Overall | 2099 | 128.05±2.27 | 902 | 71.46±.23 |
| CV% | | 22.72 | | 51.8 |
| Parity | | NS | | NS |
| Parity 1 | 517 | 125.63±1.03 | 269 | 75.31±2.25 |
| Parity 2 | 625 | 128.18±0.94 | 271 | 76.14±2.24 |
| Parity 3 | 381 | 128.08±1.20 | 141 | 70.62±3.11 |
| Parity 4 | 238 | 126.82±1.52 | 72 | 78.09±4.35 |
| Parity 5 | 132 | 129.11±2.05 | 69 | 65.34±4.44 |
| Parity 6 | 76 | 128.14±2.70 | 29 | 63.57±6.86 |
| Parity ≥7 | 95 | 130.40±2.41 | 52 | 71.15±5.12 |
| Cooperative | | < .0001 | | 0.0004 |
| Medhaniyalem | 745 | 126.75±3,68[b] | 361 | 77.45±1.94[a] |
| Kechi | 1278 | 130.10±0.65[a] | 542 | 65.48±1.58[b] |
| Gebera | 41 | 126.75±3.60[b] | - | - |
| Birth type | | 0.0005 | | NS |
| Single | 726 | 138.13±0.87[a] | 325 | 71.70±2.04 |
| Twin | 1263 | 129.37±0.66[b] | 541 | 71.47±1.58 |
| ≥Triplet | 75 | 116.64±2.72[c] | 37 | 71.21±6.07 |
| Sex | | < .0001 | | < .0001 |
| Male | 1109 | 132.53±0.70 | 478 | 77.47±1.69[a] |
| Female | 955 | 123.56±0.76 | 425 | 65.45±1.79[b] |
| Season | | NS | | 0.06 |
| Main rainy | 717 | 124.87±0.89[b] | 383 | 72.34±1.88 |
| Small rainy | 694 | 129.56±0.92[a] | 365 | 67.16±1.93 |
| Dry season | 653 | 129.71±0.88[a] | 155 | 74.89±2.96 |
| Birth year | | < .0001 | | < .0001 |
| 2017 | 82 | 119.82±2.60[b] | - | - |
| 2018 | 412 | 123.26±1.16[ab] | 191 | 49.30±2.67[c] |
| 2019 | 617 | 127.91±0.94[a] | '331 | 64.12±2.03[b] |
| 2020 | 509 | 128.56±1.04[a] | 199 | 84.51±2.61[a] |
| 2021 | 423 | 130.51±1.14[a] | 182 | 92.51±3.5[a] |
| 2022 | 21 | 138.23±5.14[a] | - | - |
| Owners | | < .0001 | | < .0001 |
| Season*year | | 0.0048 | | < .0001 |
| Birth type*year | | < .0001 | | NS |

current result 128.05±2.27 g/day) was comparable with the report of the [7] for Doyogena sheep, but lower than the report of [20], 141.9±0.80 gram/day for Bonga sheep breed but higher than the result of Horro and Menz sheep [19]. [17] also documents similar results of 129.1±1.16 gram/day for Bonga sheep breed.

**Post weaning average daily gain (ADG3-6).** The overall average daily body weight gain from 3 months (weaning) to 6 months age was 71.46±0.23 g. The effect of site, year, and sex on ADG3-6 has been presented in Table 2 were significant (P<0.01) whereas the effect of parity, birth type, and season were found to be non-significant. The least-squares means of ADG3-6 for Medhaniyalem, and Kechi Tuta were 77.45±1.94, and 65.48±1.58 gram/day respectively. Male lambs had higher ADG3-6 than females (77.47±1.69 vs 65.45±1.79 gram/day. The

current result was higher than other indigenous Ethiopian sheep postweaning daily weight gain reported so far except the previously established CBBP sites of Doyogena and Bonga sheep breeds reported by [7, 21], which are 80.59±3.62, and 98.7±2.40 gram/day respectively. There was better daily weight gain and positive phenotypic trend for Dawuro sheep population across selection years. The result suggested that wide variability exists among the flock with respect to post weaning weight gains.

## Reproductive performance

**Litter size (LS).**   The overall least-square mean of LS obtained was 1.67±0.04 litter/ewe. The parity of ewes, year of birth, and sites had significantly (p<0.01) influence on litter size while the birth season affect litter size at p<0.05. The result indicated that the litter size was increased as parity advanced. Lower litter size was recorded in the first parity. Litter size (increased with parity because ewes were physiologically mature with ewes age [7]. Similarly, lambing year and site effect were significant effects (p<0.01) on litter size. Significantly higher litter size was observed in 2018.The current LS of 1.67±0.04 litter/ewe is much higher than several previous authors of [13, 16, 22]. However, [23] reported 1.82 litter/head/ewe which is higher than the current finding for the Moroccan D'man sheep breed.

**Lambing interval (LI).**   The overall least-square mean for LI in the current study was 239.24±31.91 days. The effect of parity on LI was significant (P<0.01). However LI was not affected by lambing year, lambing season, site, and birth type and was considered P>0.05. There was a slightly decreasing trend in LI as parity advanced. The younger ewes with parity one performed significantly (p<0.01) extended interval than her later parity. This might be attributed to the fact that they are still in their stage of growth. This result is in agreement with the report of [24] who stated that as parity increases the lambing interval decrease. In general, the result of Dawuro sheep LI is comparable with the report of [25].

**Annual reproductive rate (ARR).**   The overall least squares mean of ARR (±SE) obtained was 2.16±0.06 litter/ewe/year (Table 3). As parity increased, there was also an increase in ARR. The ANOVA showed that except the effect of parity, lambing season and lambing year were non-significant. The effect of parity on ARR was also reported by [26] that, ewes in their early parity showed a smaller ARR than ewes in the middle parities. The current ARR result was higher than the ARR of Washera sheep (1.49 ± 0.02) and local sheep around Jimma zone south western Ethiopia (1.82 ± 0.44, [26]).

**Table 3.  Least squares means ± S.E of litter size, lambing interval, and annual reproductive rate.**

| Source of Variation | Litter size | | LI(days) | | ARR | |
|---|---|---|---|---|---|---|
| | N | LSM ±SE | N | Mean ± SE | N | mean ±SE |
| Overall | 2309 | 1.67±0.04 | 518 | 239.24±31.91 | 647 | 2.19±0.06 |
| CV% | | 32.91 | | 35.34 | | 44.84 |
| Dam parity | | < .0001 | | 0.04 | | 0.000 |
| Parity 1 | 517 | 1.44±0.04[b] | 131 | 309.27±11.39[a] | 139 | 1.83±0.08[c] |
| Parity 2 | 663 | 1.61±0.04[a] | 142 | 274.23±9.24[ab] | 164 | 2.01±0.07[bc] |
| Parity 3 | 452 | 1.66±0.04[a] | 146 | 268.52±11.68[ab] | 134 | 2.22±0.08[b] |
| Parity 4 | 316 | 1.72±0.04[a] | 41 | 267.22±14.19[ab] | 86 | 2.12±0.10[b] |
| Parity 5 | 170 | 1.74±0.05[a] | 27 | 257.89±18.85[ab] | 55 | 2.73±0.13[a] |
| Parity 6 | 99 | 1.73±0.06[a] | 13 | 268.31±21.22[ab] | 33 | 2.07±0.17[bc] |
| Parity≥7 | 92 | 1.76±0.06[a] | 18 | 215.73±24.77[b] | 36 | 2.36±0.16[ab] |
| Sites | | < .0001 | | NS | | 0.000 |
| Medhaniyalem | 1036 | 1.44±0.04[b] | 285 | 309.00±6.88 | 328 | 1.88±0.05[b] |

(*Continued*)

**Table 3.** (Continued)

| Source of Variation | Litter size | | LI(days) | | ARR | |
|---|---|---|---|---|---|---|
| | N | LSM ±SE | N | Mean ± SE | N | mean ±SE |
| Kechi | 1150 | 1.74±0.04[a] | 230 | 285.30±6.80 | 286 | 2.26±0.05[a] |
| Gebera | 123 | 1.82±0.06[a] | 8 | 181.92±67.10 | 33 | 2.44±0.17[a] |
| Birth type | - | - | - | NS | - | - |
| Single | - | - | 250 | 258.73±7.35 | 250 | - |
| Twin | - | - | 255 | 268.34±7.27 | 255 | - |
| ≥triplet | - | - | 13 | 270.57±32.23 | 13 | - |
| Lambing season | | 0.03 | | NS | | NS |
| Main rainy | 704 | 1.67±0.09[b] | 190 | 274.66±8.43 | 224 | 2.09±0.06 |
| Small shower | 863 | 1.61±0.04[b] | 163 | 250.64±9.10 | 229 | 2.26±0.06 |
| Dry season | 742 | 1.73±0.03[a] | 165 | 272.34±9.04 | 194 | 2.22±0.06 |
| Lambing year | - | < .0001 | | < .0001 | | 0.000 |
| 2017 | 102 | 1.30±0.08[ab] | - | - | 54 | 1.85±0.13[b] |
| 2018 | 595 | 1.79±0.03[a] | 93 | 177.13±12.05[b] | 220 | 2.30±0.05[a] |
| 2019 | 469 | 1.83±0.03[a] | 167 | 240.74±8.99[a] | 94 | 2.35±0.02[a] |
| 2020 | 424 | 1.65±0.03[b] | 97 | 308.39±11.80[a] | 163 | 1.88±0.06[b] |
| 2021 | 554 | 1.67±0.02[b] | 150 | 278.66±90.49[a] | 116 | 2.59±0.09[a] |
| 2022 | 165 | 1.76±0.17[a] | 11 | 324.49±35.04[a] | - | - |
| Owners | | < .0001 | | < .0001 | | < .0001 |
| Season*year | | 0.003 | | < .0001 | | ** |

## Conclusions

The present study provided some useful information for the progress of phenotypic performance of productive and reproductive traits of Dawuro sheep across selection years. There were variations among values of growth and reproductive performance traits that indicated the presence of within-breed variation, and there was a good opportunity to continue the selection program or scale out to new sites. Therefore, existing sites should be strengthened for further genetic improvement. Promising results of selection were notified from the ongoing Dawuro sheep CBBP: Weaning weight, six months, and litter size traits were comparable with other known indigenous sheep breeds of Ethiopia. The current growth and reproductive traits results indicated that Dawuro sheep could be improved through a continuous selective breeding program.

## Acknowledgments

The author are thankful to smallholder participant farmers of the study sites for participating in keeping and handling their sheep while collecting performance data collections. We also acknowledge Dawro zone Tocha and Kechi Tuta districts professionals, and enumerators for their immense contribution in data collection.

## Author Contributions

**Conceptualization:** Kebede Habtegiorgis Beshah.

**Data curation:** Kebede Habtegiorgis Beshah.

**Formal analysis:** Kebede Habtegiorgis Beshah.

**Funding acquisition:** Ayele Abebe.

**Investigation:** Kebede Habtegiorgis Beshah.

**Methodology:** Kebede Habtegiorgis Beshah.

**Project administration:** Ayele Abebe.

**Resources:** Adisu Jimma.

**Supervision:** Kebede Habtegiorgis Beshah, Adisu Jimma, Deribe Gemiyo, Ayele Abebe.

**Validation:** Ayele Abebe.

**Writing – original draft:** Kebede Habtegiorgis Beshah.

**Writing – review & editing:** Kebede Habtegiorgis Beshah.

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
