## [Decision Letter · Decision Letter 0]

4 Oct 2022

PONE-D-22-25221Productive and reproductive performance of Dawuro highland sheep managed under community-based breeding program in EthiopiaPLOS ONE

Dear Dr. Kebede Habtegiorgis,

Thank you for submitting your manuscript to PLOS ONE. After careful consideration, we feel that it has merit but does not fully meet PLOS ONE’s publication criteria as it currently stands. Therefore, we invite you to submit a revised version of the manuscript that addresses the points raised during the review process.

We look forward to receiving your revised manuscript.

Kind regards,

Carlos Alberto Zúniga-González, Ph.D

Academic Editor

PLOS ONE

“The  authors  are  thankful  to  the  Ethiopian Agricultural  Research  Institute  (EARI), Agricultural growth program(AGP I & II)  and  Southern Agricultural Research Institute, Areka agricultural research center  for  financing  the  research. We acknowledge Dawro zone Tocha and Kechi Tuta districts professionals, Data enumerators for their immense contribution in data collection.”

“This study was supported by Ethiopian Agricultural  Research  Institute  (EARI), Agricultural growth program(AGP I & II), Southern Agricultural Research Institute, Areka agricultural research center and Dawuro zone Tocha district livestock offices.”

6. Thank you for stating the following in your Competing Interests section: 

“The authors declare that they have no conflict of interest.”

 This information should be included in your cover letter; we will change the online submission form on your behalf."

8. We note that Figure 1 in your submission contain [map/satellite] images which may be copyrighted. All PLOS content is published under the Creative Commons Attribution License (CC BY 4.0), which means that the manuscript, images, and Supporting Information files will be freely available online, and any third party is permitted to access, download, copy, distribute, and use these materials in any way, even commercially, with proper attribution. For these reasons, we cannot publish previously copyrighted maps or satellite images created using proprietary data, such as Google software (Google Maps, Street View, and Earth). For more information, see our copyright guidelines: http://journals.plos.org/plosone/s/licenses-and-copyright.

Additional Editor Comments :

Dear authors I have reviewed the comments of the reviewers and I consider it very important that they make the observations indicated by the reviewers, I agree with the points for improvement in the abstract, the clarification of the objective defined in their research. The topic is very interesting and contributes to science in this area. However, the manuscript needs you to make the improvements. I hope that they can be concluded soon to continue with the editorial process.

Reviewers' comments:

Reviewer's Responses to Questions

**Comments to the Author**

1. Is the manuscript technically sound, and do the data support the conclusions?

Reviewer #1: Yes

Reviewer #2: Partly

2. Has the statistical analysis been performed appropriately and rigorously? 

Reviewer #1: Yes

Reviewer #2: Yes

3. Have the authors made all data underlying the findings in their manuscript fully available?

Reviewer #1: Yes

Reviewer #2: Yes

4. Is the manuscript presented in an intelligible fashion and written in standard English?

Reviewer #1: No

Reviewer #2: No

5. Review Comments to the Author

Reviewer #1: L57: complete the braxket

L60; delete:" (Agricultural office of Dawuro zone). "

L77: Taken from or taken at?

L80: what is BW?

L100-125: why animal owner is not used as a random effect? dont you think that owner to owner managemnent practices varied that might have significantly affected the trait of interest? I feel authors should try one extra model considering animal owner as a effect and see, if any variance in=s contributed by this.

L129: least square: least squares

L138: single. Twin : replace . with ,

L138-139: rephrase

L146 there is no initiating (

L162; what is "select n with"

Q1: what was the maximum number of parities for this data?

Q2: every time you write least square: make it 'least squares'

L183: grams repeated twice

L189: complete bracket

Q3: Did you record the pedigree information of the aniamls recorded in this data?

Q4: What was the accuracy of recoirding of teh phenotypic data? whether animals were weighed on exactly the target age?

Q5: How did you account for. different feeding regime adopted by different owners?

Reviewer #2: Review on

Productive and reproductive performance of highland Dawuro sheep managed under community-based breeding program in Ethiopia

Decision: Major revision

I have carefully reviewed the manuscript number (PONE-D-22-25221) titled " Productive and reproductive performance of highland Dawuro sheep managed under community-based breeding program in Ethiopia”. After careful reading, I show great interest. The study aimed to evaluate the early growth and reproductive performance of Dawuro sheep managed under a community-based breeding program. The topic is very interesting and worthy of investigation. However, the manuscript needs the following major revision before publication. The detailed comments are as follows:

1) The abstract is needs to be improved. Author should report results precisely, add some quantitative results in abstract. At the end of the abstract, readers would like to see some practical policy proposal rather than a general policy statement. Please propose some solid policy implications derived solely from your own results.

2) The Introduction fails to motivate the study. In the present form, it resembles a mini-review of literature, rather than discussing any policy-level problem. Why this study is necessary? What policy level problem this study is addressing? How the study is expected to provide any solution to that problem? How does the choice of sample is complementing that problem? Are the results and policies generalizable? The introduction is silent in all these aspects. The author should improve introduction section and explain generic information.

3) Contribution of study is missing. Authors should report the contribution in the second last paragraph of the introduction section. I suggest selecting the most convincing single contribution and explaining and emphasizing it in more detail.

4) What is the aim of the review of literature? The authors have merely listed out the studies without even creating a debate among them. Without that debate and thoughtful contradictions, the research gap cannot be substantiated. There are inconsistency and lack of coherence between paragraphs. The sections/paragraphs should derive from one to another, particularly in sub-section of literature.

5) The literature review section is short; it is recommended that author should add more literature and add

6) The results are presented not in good order; however, some more in-depth discussion of your main findings would be fruitful. The author(s) need to compare their results (each finding) with past studies (what was provided in the article is not compares of results but an explanation of views from past authors) and in comparing the result from the empirical investigations the author(s) should as much as possible provide a recast of the comparison made and the supposed implications or advantages of the new finding made with those discovered by past authors. This will ensure justice to the extant literature and evincing the superiority of the current findings over the past findings.

7) Please include nomenclature of acronyms/abbreviations/symbols used in your manuscript and position it just after the last section.

8) The overall language of manuscript is not good. Author should need to improve the language of paper. In summary, after consideration of the above-mentioned concerns, the manuscript would be suggestable for publication.

6. PLOS authors have the option to publish the peer review history of their article (what does this mean?). If published, this will include your full peer review and any attached files.

Reviewer #1: No

Reviewer #2: No

---

## [Author Response · Author response to Decision Letter 0]

4 Feb 2023

1. funding statament was removed from the manuscript 

2. The data underlying the results presented in the study are available from the Ethiopian Institute of agricultural research (EIAR), Ethiopia, Addis Ababa, contact at www.eiar.gov.et or Director, Livestock Research Directorate, pone number: 0116-45-44-32/0913380858/0116457412., email: livestock.research@eiar.gov.et. In addition, data are available from the southern agricultural research institute(www.sari.gov.et) livestock research directorate, South Ethiopia, Hawasa via email livstokresearch@sari.gov.et/ or directly from the corresponding author via email Kebede.habtegiorgis@gmail.com

---

## [Decision Letter · Decision Letter 1]

28 Mar 2023

Productive and reproductive performance of Dawuro highland sheep managed under community-based breeding program in Ethiopia

PONE-D-22-25221R1

Dear Dr. Kebede Habtegiorgis Habtegiorgis,

We’re pleased to inform you that your manuscript has been judged scientifically suitable for publication and will be formally accepted for publication once it meets all outstanding technical requirements.

Kind regards,

Carlos Alberto Zúniga-González, Ph.D

Academic Editor

PLOS ONE

Additional Editor Comments (optional):

Dear author, thanks for your improvements. I have checked that all observations were incorporated. My sincere congratulation, my decision is accepted.

Reviewers' comments:

Reviewer's Responses to Questions

**Comments to the Author**

1. If the authors have adequately addressed your comments raised in a previous round of review and you feel that this manuscript is now acceptable for publication, you may indicate that here to bypass the “Comments to the Author” section, enter your conflict of interest statement in the “Confidential to Editor” section, and submit your "Accept" recommendation.

Reviewer #1: (No Response)

2. Is the manuscript technically sound, and do the data support the conclusions?

Reviewer #1: Partly

3. Has the statistical analysis been performed appropriately and rigorously? 

Reviewer #1: N/A

4. Have the authors made all data underlying the findings in their manuscript fully available?

Reviewer #1: (No Response)

5. Is the manuscript presented in an intelligible fashion and written in standard English?

Reviewer #1: (No Response)

6. Review Comments to the Author

Reviewer #1: Authors have not responded point by point to the queries raised by me.

I would need a response to all teh quesries raised and mention their revised line number wherever required in teh revised manuscript.

7. PLOS authors have the option to publish the peer review history of their article (what does this mean?). If published, this will include your full peer review and any attached files.

Reviewer #1: **Yes: **g r gowane

---

## [Editor Report · Acceptance letter]

11 Apr 2023

PONE-D-22-25221R1 

Productive and reproductive performance of Dawuro highland sheep managed under community-based breeding program in Ethiopia 

Dear Dr. Beshah:

I'm pleased to inform you that your manuscript has been deemed suitable for publication in PLOS ONE. Congratulations! Your manuscript is now with our production department. 

Kind regards, 

on behalf of

Dr. Prof. Carlos Alberto Zúniga-González 

Academic Editor

PLOS ONE